# Cardiac Sympathetic Activity and Rhythm Control Following Pulmonary Vein Isolation in Patients with Paroxysmal Atrial Fibrillation—A Prospective ^123^I-mIBG-SPECT/CT Imaging Study

**DOI:** 10.3390/jpm11100995

**Published:** 2021-09-30

**Authors:** Philipp S. Lange, Christian Wenning, Nemanja Avramovic, Patrick Leitz, Robert Larbig, Gerrit Frommeyer, Michael Schäfers, Lars Eckardt

**Affiliations:** 1Department of Cardiology II (Electrophysiology), University Hospital Münster, Albert-Schweitzer-Campus 1, D-48149 Münster, Germany; patrick.leitz@ukmuenster.de (P.L.); robert.larbig@mariahilf.de (R.L.); gerrit.frommeyer@ukmuenster.de (G.F.); lars.eckardt@ukmuenster.de (L.E.); 2Department of Nuclear Medicine, University Hospital Münster, D-48149 Münster, Germany; Christian.Wenning@hospital-lingen.de (C.W.); Nemanja.Avramovic@glkn.de (N.A.); Michael.schaefers@ukmuenster.de (M.S.); 3European Institute for Molecular Imaging, University of Münster, D-48149 Münster, Germany

**Keywords:** sympathetic innervation, atrial fibrillation, pulmonary vein isolation, mIBG

## Abstract

Background: Pulmonary vein isolation (PVI) and antiarrhythmic drug therapy are established treatment strategies to preserve sinus rhythm in atrial fibrillation (AF). However, the efficacy of both interventional and pharmaceutical therapy is still limited. Solid evidence suggests an important role of the cardiac sympathetic nervous system in AF. In this blinded, prospective observational study, we studied left ventricular cardiac sympathetic activity in patients treated with PVI and with antiarrhythmic drugs. Prospectively, Iodine-123-benzyl-guanidine single photon emission computer tomography (^123^I-mIBG-SPECT) was performed in a total of 23 patients with paroxysmal AF, who underwent PVI (*n* = 20) or received antiarrhythmic drug therapy only (*n* = 3), respectively. ^123^I-mIBG planar and SPECT/CT scans were performed before and 4 to 8 weeks after PVI (or initiation of drug therapy, respectively). For semiquantitative SPECT image analysis, attenuation-corrected early/late images were analyzed. Quantitative SPECT analysis was performed using the AHA 17-segment model of the left ventricle. Results: PVI with point-by-point radiofrequency ablation led to a significantly (*p* < 0.05) higher visual sympathetic innervation defect score when comparing pre-and post PVI. Newly emerging innervation deficits post PVI were localized predominantly in the inferior lateral wall. These findings were corroborated by semiquantitative SPECT analysis identifying inferolateral segments with a reduced tracer uptake in comparison to SPECT before PVI. Following PVI, patients with an AF relapse showed a different sympathetic innervation pattern compared to patients with sufficient rhythm control. Conclusions: PVI results in novel defects of cardiac sympathetic innervation. Differences in cardiac sympathetic innervation remodelling following PVI suggest an important role of the cardiac autonomous nervous system in the maintenance of sinus rhythm following PVI.

## 1. Background

Atrial fibrillation (AF) is the most common arrhythmia affecting 2.5–3.2% of people worldwide [1]. AF is associated with a significant increase in stroke and heart failure and a doubling of risk for dementia. Besides antiarrhythmic pharmaceutical therapy, pulmonary vein isolation (PVI) by catheter ablation is the key therapy for rhythm control of AF. However, despite significant technical advances in PVI ablation strategies, long-term success rates after ablation still remain limited and grossly unpredictable. Strategies to improve ablation success have not only focused on the durability of pulmonary vein isolation lesions [2,3] but also on additional ablation of sympathetic ganglia and plexus [4]. In small scale clinical studies, ablation of cardiac sympathetic ganglia has yielded beneficial results [5,6,7], but the role of autonomous cardiac sympathetic innervation in AF is not yet fully understood [8,9].

Several imaging studies have provided first insights into the effects of PVI on cardiac sympathetic innervation. Arimoto et al. analysed global parameters from planar imaging (heart/mediastinum (H/M) ratio and washout-rate (WR)) and demonstrated that a high WR of ^123^I-mIBG (scan performed 5 days after PVI) was an independent predictor of AF relapses following PVI [10]. Lower H/M ratios were found in patients suffering from recurrences. Masuda et al. found that PVI restored sympathetic innervation via attenuation of excessive adrenergic tone in HF patients. Elevated sympathetic nervous tone three months post-ablation was demonstrated to be a reliable predictor of AF recurrence [11]. In a consecutive series of 16 patients, our group has recently demonstrated first evidence of regional cardiac sympathetic denervation after PVI [12]. To corroborate these findings, we initiated a prospective blinded observational imaging study using ^123^Iodine-metaiodobenzylguanidine (^123^I-mIBG) SPECT/CT pre- and post PVI. 

## 2. Methods

### 2.1. Patients

In this prospective, blinded observational trial, a total of 29 patients were included. ^123^I-mIBG-SPECT/CT scans were performed and analyzed in patients with paroxysmal AF, who underwent PVI. ^123^I-mIBG-SPECT/CT was performed before and 4 to 8 weeks after PVI or initiation of drug therapy, respectively. Patients were between 35 and 70 years old and received standard antiarrhythmic therapy including beta-blockers, and flecainide (patients characteristics see Table 1). Patients with structural heart disease, impaired left ventricular systolic function, and/or coronary heart disease were excluded from the study. 

Three patients did not undergo RF ablation and received antiarrhythmic drug treatment only (subgroup “AAT only”). 

One patient dropped out of the study due to claustrophobia, one SPECT/CT could not be analysed for technical reasons, 4 patients received one SPECT/CT only. Therefore, a complete imaging data set (SPECT/CT before and after treatment) was available in 23 patients (Table 1). 

Patients received a systematic follow up in our outpatient clinic with repetitive Holter recordings and by telephone interview for a total of six months following PVI.

### 2.2. Radionuclide (^123^I-mIBG) Imaging of Cardiac Sympathetic Innervation

Radionuclide (^123^I-mIBG) imaging was performed as described previously [12]. In brief, before intravenous administration of ^123^I-mIBG, the uptake of free ^123^I was blocked by oral administration of potassium perchlorate (500 mg) 30 min prior to the ^123^I-mIBG injection. According to a proposal for standardization of ^123^I-mIBG cardiac sympathetic imaging by the EANM Cardiovascular Committee and the European Council of Nuclear Cardiology [13] all patients received 370 MBq of ^123^I-mIBG (AdreView, GE Healthcare) and underwent anterior planar imaging of the thorax and a SPECT/CT (low-dose computed tomography for attenuation correction) of the heart at 15 min (early) and four hours (late) post injection. Data acquisition was performed on a hybrid 2-slice SPECT/CT device (Symbia T2, Siemens Medical Solutions, Malvern, PA, USA). Emission data were acquired with parallel-hole, medium-energy, high resolution collimators, the two detector heads positioned in a 90° angle, with a 20% symmetric energy window centered at 159 keV. Further acquisition parameters were: 32 rotation steps, 2.8° rotation per stop, 90° each head, and 25 seconds per rotational projection. All SPECT data was acquired ECG-gated (8 ECG gates). Left ventricular endsystolic and enddiastolic volumes (ESV, EDV) and left ventricular ejection fraction (LVEF) were automatically calculated from the ECG-gated SPECT of the late (4 h p.i.) study by means of the Corridor 4D-MSPECT software package (version 5.1, INVIA Medical Imaging Solutions, Ann Arbor, Michigan, USA). CT scans were acquired during free breathing without ECG-gating, spiral mode with pitch 1.4, tube voltage 130 kV, 30 mAs, scan time 14.67 s, CTDI (computed tomography dose index) approx. 3.2 and DLP (dose-length product) 55. CT images were reconstructed at a 5.0 mm thickness by using a reconstruction algorithm with a 512 × 512 matrix and a full-chest-size-adapted FOV of 50 × 50 cm.

### 2.3. Analysis

Analysis of planar images and semiquantitative and quantitative SPECT image analysis were performed by two experienced nuclear medicine specialists blinded for the procedure. Analysis of planar images and SPECT analysis was performed as described previously [12].

Analysis of planar images. H/M ratio was determined from the counts/pixel in a visually drawn heart region of interest (ROI) divided by the counts/pixel in a visually drawn mediastinum ROI in the mid-line upper chest positioned to reflect the region with lowest background activity, for the early images as well as for the late images. The global washout-rate (WR) was calculated as described elsewere [12].

SPECT analysis. For semiquantitative SPECT image analysis attenuation-corrected and uncorrected early/late images were analyzed using the Corridor 4D-MSPECT software package (version 5.1, INVIA Medical Imaging Solutions, Ann Arbor, MI, USA). Images were resliced into short and horizontal/vertical long axis slices for clinical reading. Additionally, bulls eye plots and a 17-segment model of the left ventricle were calculated for each SPECT images set. All images were processed by certified nuclear medicine technologists and interpreted by two experienced and independent nuclear cardiologists blinded to the clinical data. Each reader scored all SPECT image sets: relative regional tracer uptake in relation to the maximum regional myocardial ^123^I-mIBG uptake was classified using a 17-segment model of the left ventricle and a semiquantitative five-point scale (0 = normal uptake, 1 = mildly reduced uptake, 2 = moderately reduced uptake, 3 = severely reduced uptake and 4 = absent uptake), as previously described [14,15]. A “summed defect score” (SDS) was calculated as the sum of all segmental defect scores in the late study (4 h p.i.). A SDS ≥ 3 was considered to be an innervation deficit. For quantitative SPECT analysis, a 17-segment model of the left ventricle was calculated by means of a contour finding algorithm, developed and validated by our group and previously published (ESM, elastic surface model) [16]. Left ventricular segmentation was performed automatically. Segmental ^123^I-mIBG washout was calculated as the difference of the percentage ^123^I-mIBG uptakes between the early (15 minutes p.i.) and late (4 h p.i.) SPECT images. Segmental left ventricular ^123^I-mIBG uptake and washout (before PVI vs. after PVI) were compared statistically.

### 2.4. Pulmonary Vein Isolation

PVI was carried out as described previously [12]. Briefly, two 6-F sheaths were placed in the left groin. A decapolar steerable catheter (St Jude Medical, Inc, St Paul, MN, USA or Inquiry, IBI, Irvine Biomedical, Inc, Irvine, CA, USA) was placed in the coronary sinus via one of the sheaths. Two 8-F sheaths were placed in the right groin, one being an SL1-sheath (Daig SL1, Abbott, Inc, North Chicago, IL, USA) for transseptal puncture. Transseptal puncture was performed with a transseptal needle (BRK, Abbott, Inc, North Chicago, Illinois, USA). The sheath was subsequently replaced by a steerable 8.5-F sheath (AGILIS NxT, Abbott, Inc, North Chicago, IL, USA). A second transseptal puncture was performed with the SL1 sheath. Both long sheaths were flushed continuously with heparinized solution (1000 U/1 L NaCl). A total of 5000–7500 IUs of heparin were administered following the second transseptal puncture. Activated clotting time was subsequently measured every 30 minutes and maintained above 250 seconds throughout the ablation procedure. Thereafter, selective angiography of the PVs was performed with about 40 mL of nonionic contrast (Ultravist 370, Bayer, Germany). A 3D geometry of the left atrium was acquired using the Ensite NavX Velocity (Abbott). Thereafter, an antral point-by-point circumferential ablation around ipsilateral PVs, with a distance of 0.5 to 1.0 cm from the ostia, using a 4-mm open-tip irrigated catheter (IBI Therapy Coolpath Duo 7-Fr, Abbott) was performed. Maximum power was set to 30 W, going selectively up to 40 W if PV isolation could not be achieved, especially at the anterior ridge border of the lateral PVs. Temperature was limited to 43 °C. Irrigation was adjusted manually between 17 and 30 mL/min. Complete electrical isolation was monitored and confirmed by the decapolar catheter during sinus rhythm and differential pacing maneuvers. Electrical cardioversion was performed if the patient remained in AF after PV isolation. 

### 2.5. Statistics

Values are expressed as mean ± standard deviation. Differences in mean values were compared by either a Wilcoxon-Test for paired samples, or a Mann-Whitney-U-Test for unpaired samples, respectively. A *p* < 0.05 indicated a statistically significant difference. All statistical analyses were performed using a commercially available software package (SPSS, version 25.0, SPSS Inc., Chicago, IL, USA).

## 3. Results

### 3.1. Cardiac Sympathetic Activity before and after PVI

#### 3.1.1. Analysis of Global Sympathetic Cardiac Innervation

Mean late global heart to mediastinum (H/M) ratios did not change significantly from baseline to the follow-up scan after PVI (pre: 2.8 ± 0.5 vs. post: 2.8 ± 0.4; *p* > 0.05). Likewise, no significant change in the global washout rates was observed when comparing the baseline to the follow-up scan (PVI pre: 38.0 ± 9.0% vs. post: 36.4 ± 7.3%; *p* > 0.05; Figure 1). 

#### 3.1.2. Semiquantitative Analysis of Regional Sympathetic Cardiac Innervation

At baseline, relevant regional innervation defects in the early (15 min. p.i.) and late (4 h p.i.) SPECT images were observed in a single patient. In this case, the sympathetic innervation defect was located in the inferolateral left ventricular wall. In 3 other patients, defects were observed in the late SPECT images (apical and inferolateral).

After PVI, regional deficits were present in 4 patients in the early but in 12 patients in the late images including the patients with pre-existing deficits. Newly emerging defects post PVI were again predominantly localized in the inferolateral and the apical posterior wall. 

Semiquantitative analysis of all subjects (Figure 2) showed a significant increase in the mean summed defect score (4 h p.i.) from baseline to post PVI (pre: 2.6 ± 3.2 vs. post: 5.0 ± 4.4; *p* < 0.05). 

#### 3.1.3. Quantitative Analysis of Regional Sympathetic Cardiac Innervation

In order to study regional sympathetic cardiac innervation, quantitative SPECT data obtained from each patient before and after PVI were analysed and compared pairwise. The quantitative SPECT analysis comparing the late (4 h p.i.) ^123^I-mIBG uptake before and after PVI in each segment in all PVI patients revealed a significant reduction of uptake 4 h p.i. in the apical lateral, inferior, and inferolateral left ventricular segments 4, 5, 10, 11 and 16 after PVI (Figure 3). However, the changes in the regional washout rate did not reach the level of statistical difference in any of the segments in this analysis. 

### 3.2. Innervation Defects and AF Relapses

During follow-up, 12 patients had a recurrence of AF (*n* = 10 of these with a complete data set). In this patient subgroup, tracer uptake and washout *pre* and *post* PVI were analysed in each segment separately. The comparison *pre* and *post* PVI revealed that tracer uptake was significantly reduced in the segments 5, 11, 13 and 16 (=lateral and anterior apical) following PVI. The washout rate, however, did not differ significantly in any of the segments (Figure 4A). On the contrary, 11 patients did not display an AF relapse during follow up (*n* = 10 of these with a complete data set). In the analysis of regional innervation within this group with successful PVI, a reduced uptake following PVI was observed in the segments 5 and 11 only. Interestingly, the tracer uptake was elevated in segment 1 (=anterior basal). Moreover, the analysis of the washout revealed a significantly lower washout rate following PVI in the segments 2 and 3 (=septal basal), while a statistically higher tracer washout rate following PVI was observed in segment 5 (Figure 4B).

### 3.3. Sympathetic Innervation in Patients Treated with Antiarrhythmic Drugs Only

A subset of patients (*n* = 3) was treated with antiarrhythmic drugs only (“AAT”). These patients did not display significant changes in the global cardiac sympathetic innervation determined by late H/M ratios and global washout rate (Figure 5A,B). Likewise, a significant change in the summed defect score was not observed (Figure 5C). Quantitative 17-segment analysis comparing tracer uptake before and after initiation of drug therapy did not reveal significant changes. This further supports the notion that only the PVI intervention results in regional sympathetic innervation deficits.

## 4. Discussion

Solid preclinical and clinical evidence suggests a close interplay between sympathetic cardiac innervation and AF [17]. Previous imaging studies have suggested that the clinical course of AF correlates with global cardiac sympathetic innervation. Planar cardiac ^123^I-mIBG imaging performed by Akutsu et al. [18] revealed that a reduced H/M ratio was an independent predictor for the development of permanent AF in patients with paroxysmal AF. Arimoto et al. [10] described a high global cardiac ^123^I-mIBG washout rate as an independent predictor of AF recurrence. Analysis of regional ^123^I-mIBG uptake in patients with AF was first published by our group [12]. In that retrospective analysis, newly emerging innervation deficits were observed following PVI in the inferolateral segments of the left ventricle while global ^123^I-mIBG uptake and washout remained unchanged. Thus, the present prospective study was designed to decipher the correlation between cardiac sympathetic innervation inhomogeneity and susceptibility to AF in patients following pulmonary vein isolation or initiation of antiarrhythmic drug therapy, respectively.

Cardiac sympathetic innervation has also been associated with cardiac outcome and mortality. Indeed, sympathetic remodeling has been described in structural heart diseases such as dilatative and ischemic cardiomyopathy. Since the present patients had no structural heart disease our imaging results reflect sympathetic innervation and remodeling directly associated with PVI. Interestingly, the CASTLE-AF trial [19] has demonstrated that PVI in patients with heart failure was associated with a significantly lower rate of a composite end point of death from any cause or hospitalization for worsening heart failure in comparison to medical therapy. Moreover, register studies have shown that PVI may be associated with a lower incidence of death in patients with AF even in the absence of structural heart disease [20].

The present study demonstrates that distinct changes of regional left ventricular sympathetic innervation assessed by ^123^I-mIBG SPECT/CT can be consistently reproduced in a prospective, blinded observational study design. In some patients, regional sympathetic innervation deficits were present even before treatment. These innervation deficits might be attributable to differences in the AF burden before treatment. Following PVI, a reduced uptake in the inferolateral left ventricle could be detected both in the semiquantitative and quantitative analysis. In accordance with our previous study, global parameters such as H/M ratio and washout rate did not change significantly following PVI, also confirming the imaging study results of Arimoto et al. [10]. In contrast to our previous study, however, significant regional changes in the washout rate were not observed when all patients treated with PVI were analyzed. This finding implies that local changes in sympathetic innervation following PVI affect primarily cardiac ^123^I-mIBG uptake. Local signal intensity in the late images (4 h p.i.) is assumed to reflect the integrity of presynaptic nerve terminals and uptake-1 function [21]. Therefore, the reduced tracer uptake in the inferolateral left ventricle may be interpreted as a local sympathetic innervation deficit. Targeting autonomous ganglia and sympathetic nerve fibers in close proximity to PV ostia during left atrial ablation may be a possible explanation for this finding [18,22]. Lemery et al. described left ventricular denervation effects in 5 patients following targeted ablation of ganglionated plexus [23]. Anatomical studies have described a high density of sympathetic nerve fibers and ganglionated plexus near the PV ostia and along the pulmonary veins. Neurons from these ganglionated plexus innervate the ventricles and have been shown to modulate ventricular function [18]. In fact, the observed regional sympathetic innervation defects are located mostly in the inferolateral wall of the left ventricle close to the basis of the heart. These parts of the left ventricle receive rich sympathetic innervation from neurons of ganglionated plexus located around the posterior wall of the left atrium and along the pulmonary veins.

High circulating catecholamine levels have been hypothesized to compete with ^123^I-mIBG for neuronal uptake via the norepinephrine transporter thereby leading to a decreased ^123^I-mIBG uptake [21,24]. Therefore, a locally elevated catecholamine concentration might also contribute to a locally decreased ^123^I-mIBG uptake. Interestingly, several cardiac^123^I-mIBG imaging studies performed in patients with heart failure, coronary artery disease, and inherited arrhythmias have demonstrated a decreased global cardiac ^123^I-mIBG uptake. However, in these studies, the global reduction of ^123^I-mIBG uptake was associated with an increased global washout [25,26,27]. The washout rate is thought to reflect turnover of catecholamines that relates to the degree of sympathetic drive. Hence, an isolated reduction of ^123^I-mIBG uptake in combination with an unchanged washout rate probably reflects a regionally decreased sympathetic activity due to a local effect of radiofrequency energy on sympathetic nerve fibers or sympathetic ganglia.

Since the follow up ^123^I-mIBG scan was performed several weeks after pulmonary vein isolation, it seems that the changes in cardiac sympathetic innervation following PVI lead to a persistent remodeling of cardiac sympathetic innervation. Therefore, these changes are putatively relevant for AF relapses early after PVI. Interestingly, the pattern of regional sympathetic innervation deficits differed between patients with and without AF recurrence. In addition to a reduction of uptake in the inferolateral segments, patients with AF relapse displayed a significantly reduced uptake in apical segments. Moreover, an elevated washout in the basal septal segments was observed in patients without AF relapse. Since artefacts in these segments are common, artificially reduced uptake cannot be completely ruled out. However, no significant changes in the washout rate could be detected in the group of patients with AF relapse. Our study therefore suggests that changes in the washout rate and elevated uptake in the anterior basal segment may be associated with successful ablation.

While the number of patients treated with antiarrhythmic drugs only is small, conclusions drawn from this subset of patients should be viewed with caution. However, the sympathetic innervation changes in the anterior segments appear to outpace a pure sympathetic denervation effect as a consequence of radiofrequency ablation. Indeed, they might indicate a remodeling of cardiac sympathetic innervation that accompanies successful treatment of AF. Taken together, the changes in sympathetic innervation determined by ^123^I-mIBG SPECT/CT probably reflect both direct denervation effects by direct ablation of ganglionated plexus and remodeling of cardiac and particularly left ventricular sympathetic innervation.

Taken together, the results of the present study support the notion that PVI is associated with a remodeling of cardiac sympathetic innervation. The observed innervation deficits may be the result of a direct effect of radiofrequency ablation on sympathetic ganglia and fibers in response to successful rhythm control. The observed inhomogeneity of cardiac sympathetic innervation in defined segments of the left ventricle regional may have a prognostic impact and seems to determine susceptibility to AF in individual patients with AF recurrences after PVI.

The present study has a number of important limitations. So far, we do not know whether the results of our study can be reproduced in patients with (long standing) persistent AF or in AF patients with structural heart disease. Besides sympathetic cardiac innervation, parasympathetic innervation also plays an important role in atrial fibrillation [9], and it has been proposed that both parts of the autonomic nervous system are required to initiate atrial fibrillation. In fact, modulation of parasympathetic innervation by pulmonary vein isolation cannot be ruled out.

In addition, functional measurements such as heart rate variability have not been performed. Moreover, the small patient number limits the power of the study. In particular, due to the low number of patients treated with antiarrhythmic drugs only, imaging results from this subgroup should be interpreted with caution. Finally, the low spatial resolution of SPECT/CT impedes the imaging of sympathetic nerve structures and of ganglionated plexus.

## 5. Conclusions

PVI is associated with newly emerging regional cardiac innervation defects of the inferolateral wall. The regional changes in cardiac sympathetic innervation most probably result from both concomitant ablation of sympathetic nerve plexus due to PVI and sympathetic remodelling following PVI. Significant differences in the sympathetic innervation pattern between patients without and with recurrences of AF suggest an important interplay of cardiac sympathetic innervation and PVI, and AF relapse may be predicted by cardiac ^123^I-mIBG imaging.

## Figures and Tables

**Figure 1 jpm-11-00995-f001:**
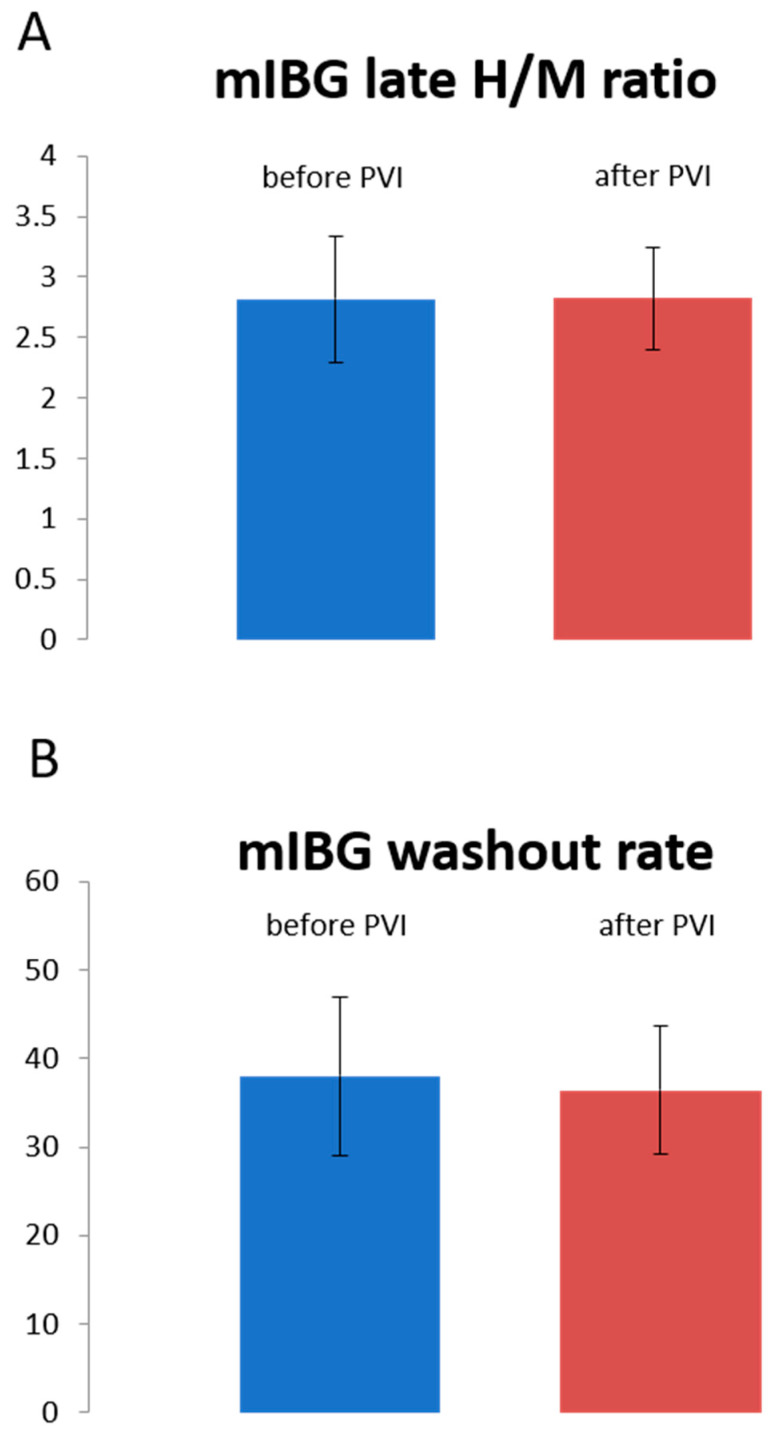
Global cardiac sympathetic innervation determined by late H/M ratios (**A**) and global washout rate (**B**). Late H/M ratios and washout rate (in %) are shown before and after PVI.

**Figure 2 jpm-11-00995-f002:**
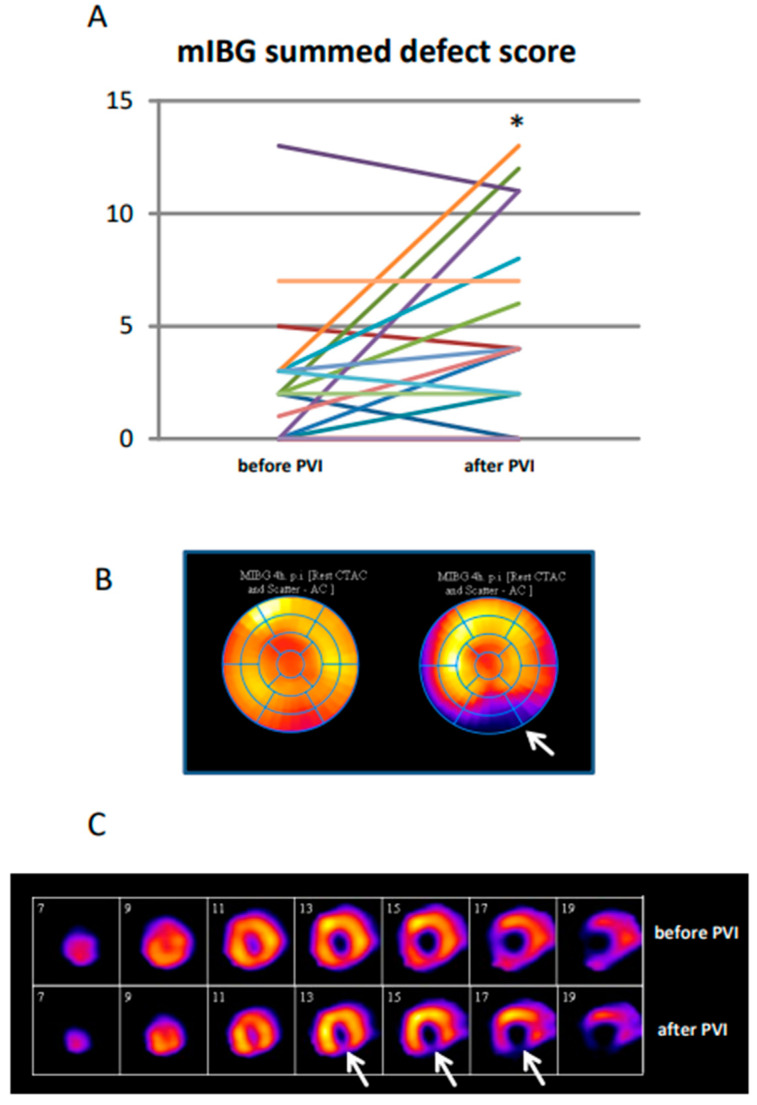
Summed defect score (SDS) in individual patients. (**A**) SDS is shown before and after PVI. * *p* < 0.05. (**B**,**C**) I-123-mIBG scan in a patient before and after PVI displaying a novel innervation deficit (white arrows) after PVI. (**B**) Polar map and (**C**) short axis slices of the left ventricle.

**Figure 3 jpm-11-00995-f003:**
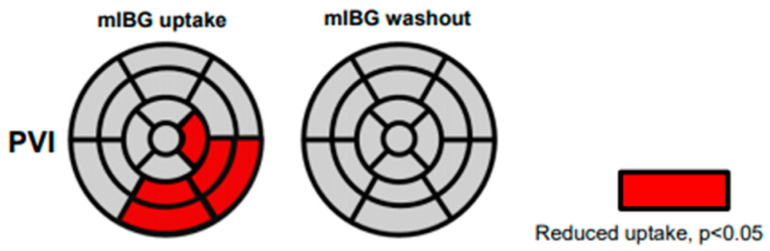
Quantitative 17-segmental analysis of the late I-123-mIBG uptake (**left**) and washout (**right**) of the left ventricle after PVI. Values are expressed as mean regional uptake in comparison to the regional maximum uptake and segmental washout (%). Significantly reduced uptake is highlighted in red (mean values after PVI compared to mean values before PVI).

**Figure 4 jpm-11-00995-f004:**
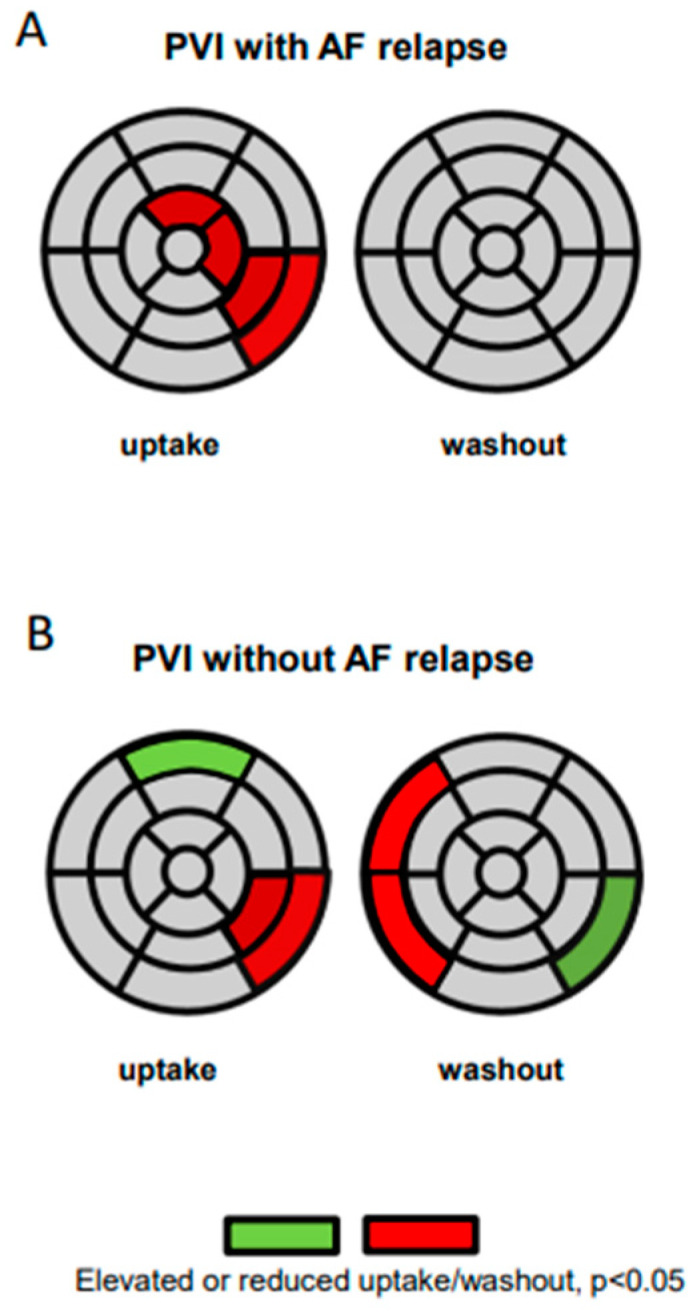
Quantitative 17-segmental analysis of the late I-123-mIBG uptake (left) and washout (right) of the left ventricle after PVI in patients with AF relapse (**A**) and in patients without AF relapse (**B**). Values are expressed as mean regional uptake in comparison to the regional maximum uptake and segmental washout (%). Significantly reduced uptake and reduced regional washout is highlighted in red; significantly elevated tracer uptake and elevated washout is highlighted in green (mean values after therapy compared to mean values before PVI).

**Figure 5 jpm-11-00995-f005:**
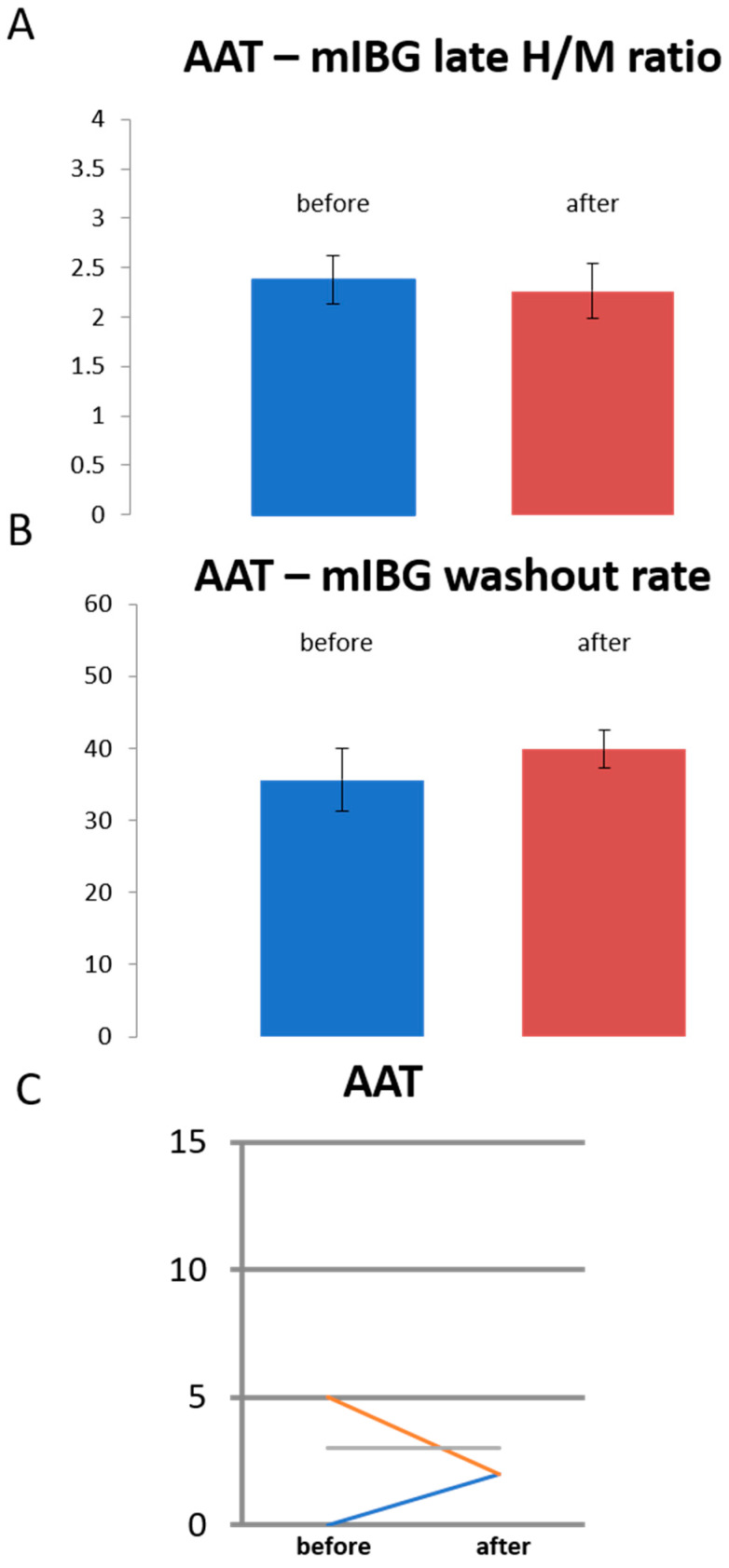
Analysis of the subgroup of patients treated with antiarrhythmic drug therapy only (“AAT”, *n* = 3). Global cardiac sympathetic innervation determined by late H/M ratios (**A**) and global washout rate (**B**). Late H/M ratios and washout rate (in %) are shown before and after initiation of drug therapy. (**C**) Summed defect score (SDS) in individual patients. SDS is shown before and after initiation of drug therapy.

**Table 1 jpm-11-00995-t001:** Individual patients characteristics.

Individual #	Treatment	Sex	Age	LA Volume (mL)	Body Mass Index	AA Drugs before Treatment (PVI or AAT Only)	Relapse after Treatment (PVI or AAT Only)	AA Drugs after Treatment (PVI or AAT Only)
1	PVI	M	55	44	25	sotalol	Yes	sotalol
2	PVI	F	58	40	21	beta blocker	Yes	beta blocker
3	AAT only	F	70	38	27	None	Yes	beta blocker, propafenone
5	PVI	M	63	119	30	beta blocker, flecainide	Yes	beta blocker, flecainide
6	PVI	M	67	87	29	beta blocker, flecainide	Yes	beta blocker, flecainide
7	PVI	F	58	60	27	beta blocker, dronedarone	None	beta blocker, dronedarone
8	AAT only	F	52	35	27	beta blocker	None	beta blocker, dronedarone
9	PVI	F	53	85	35	beta blocker, flecainide	Yes	beta blocker, propafenone
10	PVI	M	45	56	28	beta blocker, flecainide	Yes	beta blocker, flecainide
11	PVI	F	61	34	28	beta blocker, dronedarone	Yes	dronedarone
12	PVI	M	53	52	26	beta blocker, dronedarone	None	beta blocker, flecainide
13	PVI	M	62	63	31	beta blocker, flecainide	None	beta blocker, flecainide
14	PVI	M	60	39	27	beta blocker	None	beta blocker, flecainide
15	PVI	M	57	41	26	beta blocker, flecainide	Yes	beta blocker, flecainide
16	PVI	M	54	56	33	beta blocker, flecainide	None	beta blocker, flecainide
17	PVI	F	68	57	33	beta blocker, amiodarone	None	beta blocker, amiodarone
18	PVI	M	44	58	26	Beta blocker, flecainide	None	beta blocker, flecainide
19	PVI	M	48	36	29	Beta blocker, flecainide	None	Beta blocker, flecainide
20	PVI	M	51	56	34	Beta blocker, flecainide	Yes	Beta blocker, flecainide
21	PVI	M	58	71	25	flecainide	Yes	flecainide
22	PVI	M	55	72	26	Beta blocker, flecainide	Yes	Beta blocker, flecainide
23	PVI	F	58	44	22	Beta blocker, flecainide	None	none
24	PVI	F	58	49	26	Beta blocker, flecainide	None	Beta blocker, flecainide
25	AAT only	M	61	53	31	Beta blocker	None	Beta blocker, flecainide
26	PVI	M	55	42	25	Propafenone	None	Propafenone
27	PVI	M	66	152	28	Beta blocker, flecainide	Yes	Beta blocker, flecainide
28	PVI	M	53	52	24	Beta blocker	None	Beta blocker
29	PVI	M	67	87	25	Beta blocker, amiodarone	None	Beta blocker

LA, left atrium; AA, anti-arrhythmic.

## Data Availability

The datasets used and/or analysed during the current study are available from the corresponding author on reasonable request.

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
