# Peer review of "Cardiac Sympathetic Activity and Rhythm Control Following Pulmonary Vein Isolation in Patients with Paroxysmal Atrial Fibrillation—A Prospective 123I-mIBG-SPECT/CT Imaging Study"

_jpm, 2021, doi:10.3390/jpm11100995_

Round 1
Reviewer 1 Report
Dear authors,
thank you for this well presented manuscript.
One question would be if this was the first paroxysmal AF in your studied group and if not, could the number of relapses also play a role?
In the table 1 "Drug treatment before treatment" you probably mean "Drug treatment before PVI treatment" and respectively in the "after treatment box", don't you? Also in the "Relapse" you mean "Relapse after PVI", don't you?
Author Response
Dear Reviewer,
thank you very much for your important comments. Please see below our point-by-point answers to your comments and questions.
Reviewer #1
Dear authors, thank you for this well presented manuscript. One question would be if this was the first paroxysmal AF in your studied group and if not, could the number of relapses also play a role?
In our study, patients with paroxysmal atrial fibrillation were included. 123I-mIBG-SPECT/CT was performed before and 4 to 8 weeks after PVI or initiation of drug therapy, respectively.
However, the first SPECT/CT scan was not performed directly after the first diagnosis of atrial fibrillation. Therfore, the number of relapses before treatment could indeed play a role. In fact, in some patients, regional sympathetic innervation deficits were present even before treatment. These innervation deficits might be attributable to differences in the AF burden before treatment. The manuscript has been improved accordingly.
In the table 1 "Drug treatment before treatment" you probably mean "Drug treatment before PVI treatment" and respectively in the "after treatment box", don't you? Also in the "Relapse" you mean "Relapse after PVI", don't you?
The table contains data of both groups: PVI treatment ("PVI") and antiarrhythmic drug treatment only ("AAT" only). The table has been improved accordingly.
Reviewer 2 Report
The authors of the presented manuscript studied cardiac sympathetic changes in patients with paroxysmal atrial fibrillation after isolation of the pulmonary vein. At the same time, they analyzed the sympathetic activity in the heart of patients treated only with antiarrhythmic drugs. The topic of the work is interesting and brings interesting results proving the plasticity of sympathetic cardiac innervation.
I have a few remarks/questions:
- The manuscript lacks the results of functional measurements showing the degree of damage to the autonomic nervous system after PVI, e.g. HRV, which is an indicator of autonomic activity.
- The possibility of damage to the parasympathetic innervation of the heart during PVI is not mentioned in the manuscript, and therefore there is no proper discussion on this topic, e.g. in connection with the influence of both parts of the autonomic nervous system on AF.
- The legend to Figure 5 does not contain the number of measurements (patients). There were only 3 patients in this group, which is not enough to properly analyze the results and draw conclusions.
Author Response
Dear Reviewer,
thank you very much for your important remarks which we believe will substantially improve the manuscript. Please see below our point-by-point answers to your comments.
I have a few remarks/questions:
The manuscript lacks the results of functional measurements showing the degree of damage to the autonomic nervous system after PVI, e.g. HRV, which is an indicator of autonomic activity.
Thank you for your comment. Using 123I-mIBG SPECT/CT scans, we found regional cardiac sympathetic innervation deficits in this prospective study. Whether these regional innervation deficits translate into an alteration of heart rate variability has not been investigated in our study. This is an important limitation. Therefore, the „Discussion“ section has been improved accordingly.
The possibility of damage to the parasympathetic innervation of the heart during PVI is not mentioned in the manuscript, and therefore there is no proper discussion on this topic, e.g. in connection with the influence of both parts of the autonomic nervous system on AF.
The cardiac parasympathetic autonomous nervous system also plays an important role in the pathophysiology of atrial fibrillation. However, it is technically challenging to study both sympathic and parasympathic innervation simulataneously using nuclear imaging. The current study therefore focuses on sympathetic innervation. The „Discussion“ section has been improved accordingly.
The legend to Figure 5 does not contain the number of measurements (patients). There were only 3 patients in this group, which is not enough to properly analyze the results and draw conclusions.
Thank you for this important comment. Indeed, the low number of patients treated with antiarrhythmic drugs only makes ist difficult to draw valid conclusions. However, in our mind, the imaging results of this patient subgroup should not be withheld. The limited statistic power has been discussed in the „Discussion“ section of the manuscript. In order to clarify this issue, we have now improved the figure legend, and both the „Results“ and „Discussion“ section.